# Low-Profile Millimeter-Wave Metasurface-Based Antenna with Enhanced Bandwidth

**DOI:** 10.3390/mi14071403

**Published:** 2023-07-10

**Authors:** Ke Han, Yuchu Yan, Ze Yan, Chongwei Wang

**Affiliations:** 1State Key Laboratory of Information Photonics and Optical Communications, Beijing University of Posts and Telecommunications, Haidian District, Beijing 100876, China; yanyuchu@bupt.edu.cn; 2School of Electronic Engineering, Beijing University of Posts and Telecommunications, Haidian District, Beijing 100876, China; yanze2020@bupt.edu.cn (Z.Y.); chwzbj@bupt.edu.cn (C.W.)

**Keywords:** broadband antenna, characteristic mode analysis (CMA), metasurface-based antenna, millimeter-wave antenna

## Abstract

A millimeter-wave broadband metasurface-based antenna with a low profile is proposed. In order to guide the mode excitation, the characteristic mode analysis (CMA) is used for the design and optimization of the proposed antenna. Four sets of coplanar patches with different dimensions on a thin printed circuit board are used to generate four adjacent broadside modes, which are directly fed by a coaxial probe. Then, to expand low-frequency bandwidth, a new resonant mode is introduced by etching slots on the parasite patch. Meanwhile, the extra mode introduced does not significantly change the radiation performance of the original modes. Moreover, dual slots are etched on the mid patch fed by the coaxial probe, which moves the orthogonal modes of the chosen modes out of the operating band to reduce cross-polarization levels. The proposed antenna realized 25.02 % (30–38.58 GHz) impedance bandwidth with dimensions of $1.423\times 1.423\times 0.029\lambda_{0\;}^{3}$
($\lambda_{0}$ is the wavelength at 34 GHz in free space), and the realized gain in the band is 8.35–11.3 dB.

## 1. Introduction

Following the development of wireless communication technology, it has become deeply integrated into the daily life of humans. In recent years, the development of 5G has driven widespread demand for millimeter-wave technologies, which require greater bandwidth for high-speed data transmission. With the increasing use of millimeter-wave bands in wireless communications and the continuous increase in data transmission in communication scenarios, the ease of integration and the characteristics of the co-type make microstrip antennas promising for a wide range of applications. However, the disadvantage of the narrow operating band of microstrip antennas has become more and more prominent and has motivated the vigorous development of bandwidth expansion techniques for microstrip antennas.

A traditional microstrip antenna shows a narrow impedance bandwidth due to its single-resonance working mechanism. Etching slots [1] and loading short circuit pins or holes [2,3] are used to improve its performance. However, it shows low-level improvements in performance. Loading parasitic resonant patches [4,5,6] will increase cross-polarization levels and the analysis difficulty. Laminated patches [7,8] and air substrates [9] will introduce bulk to the antenna.

All the above techniques can broaden the bandwidth of the antenna well. However, these techniques make the structure of the antenna relatively complex, resulting in increasing analytical complexity. In order to implement broadband antennae with the advantages of low profile, low cost, and easy integration, metasurfaces have entered the research field. Early studies utilized composite periodic structures such as EBG (electromagnetic band-gap) and AMC (artificial magnetic conductor) to present an artificial impedance-controlled surface [10,11,12] for size reduction, wide bandwidth, and back radiation reduction. All these can be considered metasurface-based structures in a broad sense. On the one hand, metasurface is widely used in low reflection cross sections (RCS) [13,14,15,16,17] and filtering [18,19,20,21]. On the other hand, its multi-mode resonance characteristics show a prominent advantage in broadband [22,23,24,25,26,27,28], multi-band [29], and omnidirectional radiation [30,31,32,33] applications. 

In addition, metasurface antennae not only retain the characteristics of low profile and easy integration, but also have the structure of multiple radiation patches, which can achieve broadside gain and broadband bandwidth that can exceed 50% [34,35,36], so it has attracted more and more attention. In our work, these methods are developed. The periodicity of the structure is broken to achieve a better bandwidth and frequency control capability. However, a traditional metasurface requires a multi-layer PCB feeding structure, which increases the antenna profile thickness. For millimeter-wave antennas, broadband antennas with more resonant modes, low profiles, and simple structures are needed to accommodate the increasing number of millimeter-wave mobile terminals.

It is a common design method for broadband metasurface antennas to excite multiple modes of metasurface for broadband operation [22,37]. With the help of CMA, the modal characteristics of the metasurface can be predicted to a certain extent, and the effects of structural changes can be revealed. In this way, it is possible to find a way to expand the bandwidth without significantly affecting the desired modes. In order to further increase the bandwidth without changing the overall size, targeted changes can be made to the characteristics mode of the metasurface by adjusting the patch size [29], patch segmentation [38,39], cutting angle [40], patch hollowing [34], and other methods. However, the coupling between the metasurface structures is very tight, and the desired modes are inevitably affected once the metasurface structure is drastically changed. For example, the sidelobe of Modal $J_{9}$ is suppressed by splitting the corner patches into four small patches, but the sidelobe level of the H-plane is increased [29]. Moreover, the segmented small-size patch has a nearly uniform effect on the desired radiation pattern, increasing the difficulty of adjusting a particular mode. In addition, complex feeding structures may also affect the characteristic mode of the metasurface, which increases the difficulty of designing the metasurface antenna. For example, dipole feeding [41] and slot coupling feeding [35,36] introduce new modes that can be used to expand the bandwidth of the antenna, but this puts forward higher requirements for the collaborative design of metasurface and feeding structure, and it will increase the profile. 

In this paper, we propose a multi-mode resonant millimeter-wave metasurface-based antenna. Multiple patches are used to generate the characteristic broadside modes in adjacent frequency bands with coaxial probe excitation. Based on this, a new mode is introduced by etching slots on parasite patches, and an additional resonant point is added in the low-frequency region, which further enlarges the bandwidth of the antenna. Then, dual slots are etched on the mid patch fed by the coaxial probe, which moves the orthogonal modes of the chosen modes out of the operating band to reduce cross-polarization levels. Moreover, the etched slot does not significantly change the radiative properties of the original mode. Meanwhile, the metasurface-based structure consisting of multiple patches guarantees high antenna gain. Due to the simple feeding structure, the processing technique is straightforward. All this gives the antenna broadband, high gain, miniaturization, and a low profile.

## 2. Antenna Design

Figure 1 shows the configuration of the proposed antenna. The metasurface was printed on the top of a square single-layer PCB board with a thickness of 0.254 mm and a dimension of $W_{g}\times W_{g}$. The dielectric substrate was ROGERS RT/duroid 5880 ($\varepsilon_{r}=2.2$). A total of 13 patches of 4 sizes were used, which were divided into C1, C2, C3, and C4 groups. Patch C1 is in the center, on which two slits with a width of 0.15 mm are etched. All patches except C1 are square. Two groups of 4 patches of different sizes are symmetrically arranged around C1. The rest are square patches with side length $L_{4}$. The spacing between C1 and C2, C3, and C4 is $S_{12}$, $S_{13}$, and $S_{14}$, and the slots between C2 and C4, C3 and C4 is $S_{24}$ and $S_{34}$, respectively. There is a metalized via under C1, which is used to feed, and its diameter is 0.3 mm. The dimensions are summarized in Table 1.

The resonance behavior of source-free metasurface was characterized by CST. In this paper, the ground plane is infinite for CMA simulations, while in other simulations, the size of the ground plane is consistent with that of the dielectric substrate, which is $W_{g}\times W_{g}$. As shown in Figure 2, the lower surface of the dielectric substrate is set as the PEC boundary, and the remaining directions are open. When CST is used for characteristic mode analysis, the influence of feeding structure is not considered, which means CMA is carried out for metasurface without feeding structure. Five adjacent broadside characteristic modes are selected. The modal significance is shown in Figure 3, and the resonant frequencies are 28.864, 32.524, 35.656, 37.264, and 39.512 GHz, respectively. The corresponding modal current and radiation patterns at the resonant frequency are shown in Figure 4 and Figure 5. The modal current of Mode $J_{1}$ is mainly concentrated around the slots of C3 patches, which is called slot mode. The mode currents of the other four modes are concentrated in the patch sets of C1, C2, C3, and C4, respectively, which are dominated by the corresponding patch sets.

The mainly modal currents are polarized along $-45^{{}^{\circ}}$ and can be excited simultaneously by a coaxial probe. The modal significances of Modal $J_{2}$ gradually decreases after 32.5 GHz, and the contribution of the radiation mode provided by Modal $J_{2}$ to the total radiation mode of the antenna also gradually decreases. After Modal $J_{2}$ degenerates to a non-significant mode (MS ≤ 0.707), there is no other mode that can provide the required radiation mode. Therefore, the antenna gain in this frequency range will gradually decrease. Until the contribution of Modal $J_{3}$ and Modal $J_{4}$ to the total radiation pattern of the antenna cancels out the decreasing trend due to the attenuation of the radiation pattern provided by Modal $J_{2}$. In other words, the half-power bandwidth of Modal $J_{2}$ and Modal $J_{3}$ does not completely cover the frequency interval between the resonance points of the two modes, where the antenna gain deteriorates. Mode $J_{3}$ and Mode $J_{4}$ were both significant modes (MS ≥ 0.707) between their resonant frequency points, which makes these two modes present as mixed modes.

Modal $J_{5}$, which is dominated by the C4 patch group and has a resonant frequency of 39.5 GHz, is not in the operating frequency band. Modal $J_{5}$ forms a resonance point with the Modal $J_{4}$ that degenerates into an insignificant pattern. As frequencies close to 39.5 GHz, the Modal $J_{5}$ excitation level gradually increases and contributes to the dominant radiative features, which makes the antenna gain increase. Thus, the antenna gain still shows an increasing trend beyond the half-power bandwidth of the Modal $J_{4}$.

Figure 6 shows the characteristic mode analysis results of the grooveless metasurface-based antenna, which is called Antenna 2 (Figure 7b). Compared to Figure 3, the resonant frequencies of Modal $J_{2}$ and Modal $J_{4}$ change slightly after etched slots, while the radiative properties of the other modes do not change significantly. The slots etched on C3 introduce a new resonant mode, Modal $J_{1}$, in the low-frequency region, which further broadens the frequency band of the metasurface-based antenna. The etched slots on C1 cause the orthorhombic modes of the selected mode to move out of the band, optimizing the cross-polarization level. Modal currents of Modal $J_{1}$ are mainly distributed on both sides of the slots. The modal current flows around the slots on the C3 patch, consisting of two polarization currents polarized along and perpendicular to the slots, and the polarization purity of the modal current is not very high. When a coaxial probe is used for feeding, the electric field emanating from the feeding point cannot excite it to the maximum extent, and the antenna gain will be low when this mode dominates. The resonant frequency is affected by the dimensions of the slots, which can be adjusted to shift the mode into the band.

To evaluate the effectiveness of this strategy, the performance of this antenna is compared with that of two reference antennas (Figure 7). Simulated $\left|S_{11}\right|$ and broadside gain is shown in Figure 8. The slots etched on the C3 patch dominates the Modal $J_{1}$, introduces a new resonant point in the low-frequency region, and widens the impedance bandwidth of the metasurface-based antenna by 0.56 GHz. Since the polarization purity of the Modal $J_{1}$ is not high, the gain of the antenna is low with respect to the other frequency points when this mode dominates. The Modal $J_{2}$ is dominated by the C1 patch, and the etching slots essentially do not change its resonance frequency, only the impedance matching. Modal $J_{5}$, which is dominated by the C4 patches, introduces a new resonance point in the high-frequency region, further expanding the bandwidth. Since the modal currents on C4 patches are in phase with the primary polarization currents, the gain of the antenna is enhanced. When Modal $J_{3}$ and Modal $J_{4}$ dominate, the modal current on C4 patches is in phase with the main polarization current, which improves the gain deterioration in Antenna 1. However, the currents on the C1 patch are out of phase with the main polarization current, and the etched slots enhance the out-of-phase currents on the C1 patch and weaken the gain when Modal $J_{2}$ and Modal $J_{3}$ dominate. Hence the gain of Pro.Ant deteriorates again. This gain fluctuation is acceptable, considering the bandwidth gain from the etching slots.

After the introduction of the new resonance point, there is a peak impedance in the low-frequency region, as shown in Figure 9. The real part of the peak impedance increases by 10 Ω, and the reactance increases from −9.7 to −5.6. The impedance matching with the 50 Ω port is significantly improved, while the impedance of other frequencies also approaches the impedance of 50 + j0 Ω.

## 3. Parametric Study

The influences of $L_{2},\;L_{3},\;L_{4}$, and $L_{s2}$ on $\left|S_{11}\right|$ are shown in Figure 10. When Modal $J_{2}$, Modal $J_{3}$, and Modal $J_{5}$ are dominant, there is a strong polarization current across the C2 patches, so these modes are affected to varying degrees when $L_{2}$ is varied. The polarization current of Modal $J_{4}$ is mainly concentrated on the C3 patches. Since Modal $J_{4}$ is mixed to varying degrees with Modal $J_{3}$ and Modal $J_{5}$, the impedance matching of the two resonant points in the high-frequency region can be tuned by adjusting $L_{3}$, and the resonant frequency does not change significantly.

As the length of $L_{4}$ decreases, the resonant frequency of the fourth resonant point gradually shifts to a higher frequency, while the frequencies of other resonant points hardly change. As the length of $L_{s2}$ increases, so does the length of the current path flowing around the slots, and thus the resonant frequency shifts to lower frequencies. When the slot length $L_{s2}$ is shortened to 0.98 mm, the resonant frequency of the slot mode is very close to that of mode 1. If the length of $L_{s2}$ is further shortened, the resonant point generated by the slot mode can no longer be observed independently. The variation of the slot length does not affect the characteristics of the other modes.

Considering bandwidth and impedance matching, the four parameters are set as $L_{2}=2.60\;mm,\;\;L_{3}=2.53\;mm,\;\;L_{4}=2.38\;mm,\;\;{and\;L}_{s2}=1.08\;mm$. 

## 4. Measured Results

A prototype sample of the proposed metasurface-based antenna is shown in Figure 11. The metasurface-based antenna is printed by a single-layer PCB with a copper foil thickness of 0.018 mm. It is fed by a 3 × 3 × 3 ${mm}^{3}$ RF coaxial connector with a characteristic impedance of 50 Ω (the coaxial probe is not inserted into the dielectric substrate, only welded to the bottom pad). The antenna is measured by an Agilent 5230Avector network analyzer and the standard anechoic chamber, as Figure 12 shows.

Figure 13 and Figure 14 show the measured results of the antenna impedance matching, peak gain, and radiation pattern. Due to welding, thick, irregularly shaped solder is deposited at the feeding point and edges of the C1 patch at the top of the antenna, which causes a slight change in the size of the C1 patch, resulting in a difference in impedance matching and resonant frequency. The measured impedance bandwidth of the metasurface-based antenna is 25.02% (30–38.58 GHz), and the realized gain in the band is 8.35–11.3 dB. The performance of the metasurface-based antenna is in line with expectations.

As shown in Figure 15, the radiation efficiency of the antenna in the high-frequency region decreases gradually, and the in-band efficiency stays above 75%. The rapid decrease in radiation efficiency of the proposed antenna is due to the effects of the SSMP connector. Also, the effects of the measured equipment could not be completely ruled out as a possible cause.

## 5. Conclusions

In this paper, we propose a low-profile broadband metasurface-based antenna with five resonant modes. CMA is used to guide the design of the proposed antenna. A new resonant mode is introduced by etching slots, and a coaxial probe is used to excite all modes, resulting in a wide bandwidth. The etched slots expand the bandwidth without significantly affecting the original mode while reducing the cross-polarization level. The proposed antenna realized 25.02% (30–38.58 GHz) impedance bandwidth, and the realized gain in the band is 8.35–11.3 dB. The antenna possesses the advantages of having a small size, large bandwidth, and practical value.

## Figures and Tables

**Figure 1 micromachines-14-01403-f001:**
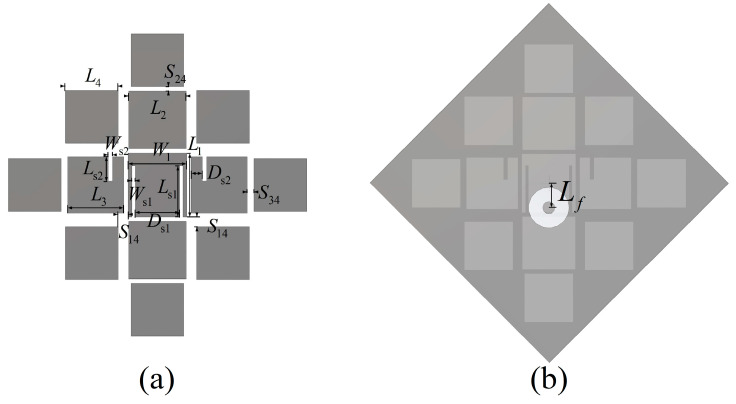
Configuration of the proposed antenna (**a**) top and (**b**) bottom.

**Figure 2 micromachines-14-01403-f002:**
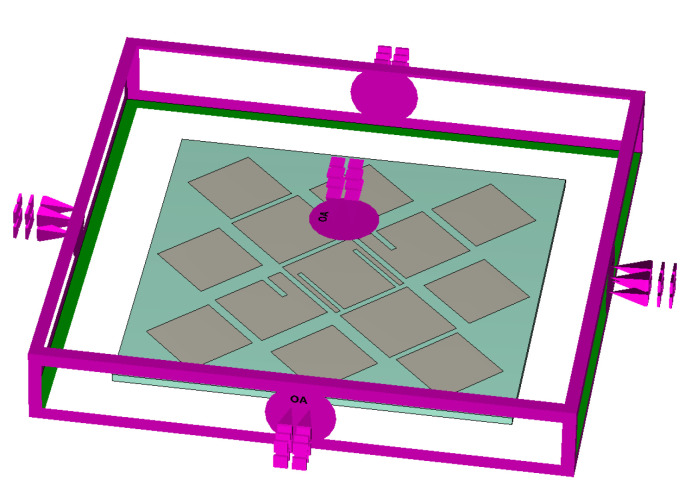
Geometrical modeling and boundary setup.

**Figure 3 micromachines-14-01403-f003:**
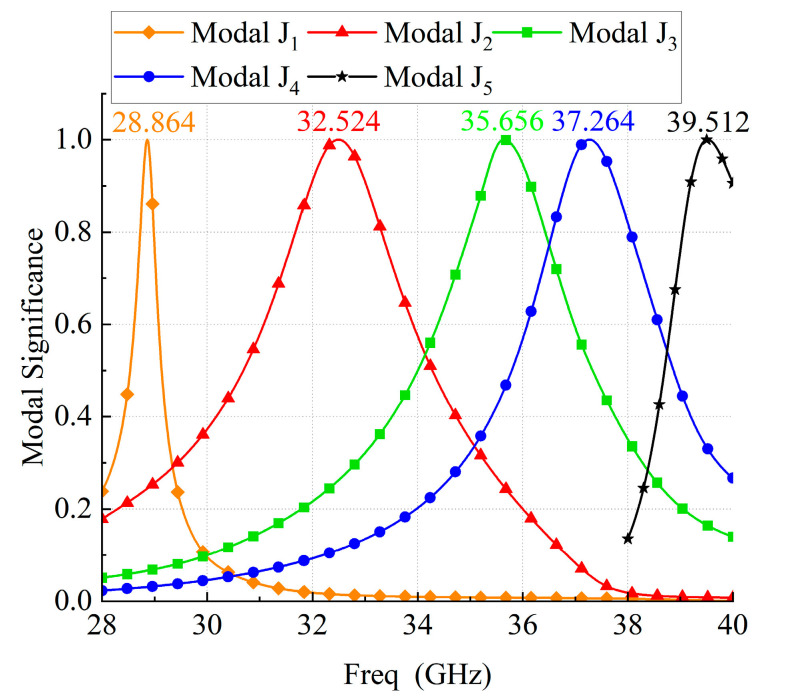
Modal significances of the proposed metasurface-based antenna.

**Figure 4 micromachines-14-01403-f004:**
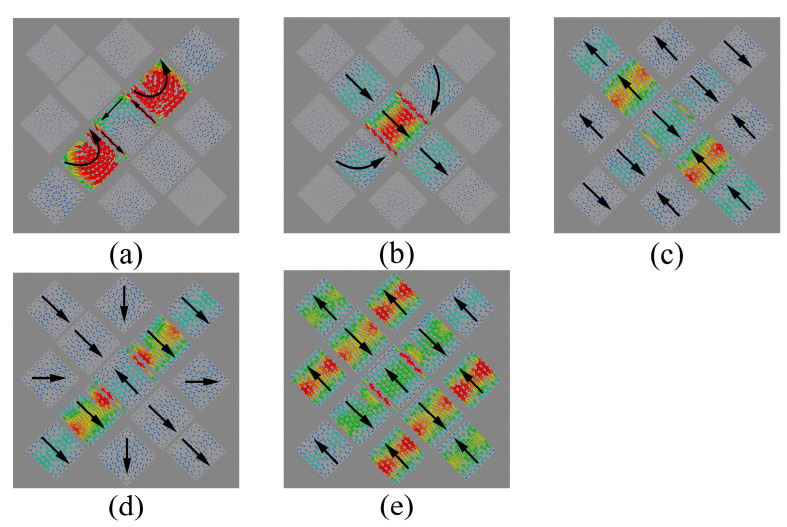
Modal currents at resonant frequency for (**a**) Modal $J_{1}$; (**b**) Modal $J_{2}$; (**c**) Modal $J_{3}$; (**d**) Modal $J_{4}$.

**Figure 5 micromachines-14-01403-f005:**
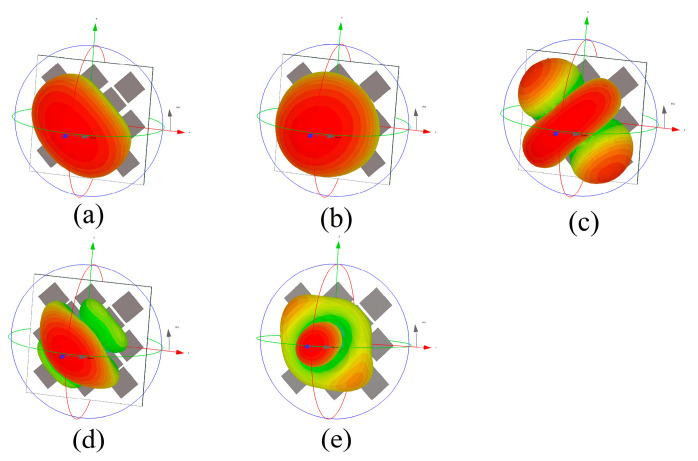
Modal radiation pattern at resonant frequency for (**a**) Modal $J_{1}$; (**b**) Modal $J_{2}$; (**c**) Modal $J_{3}$; (**d**) Modal $J_{4}$; and (**e**) Modal $J_{5}$.

**Figure 6 micromachines-14-01403-f006:**
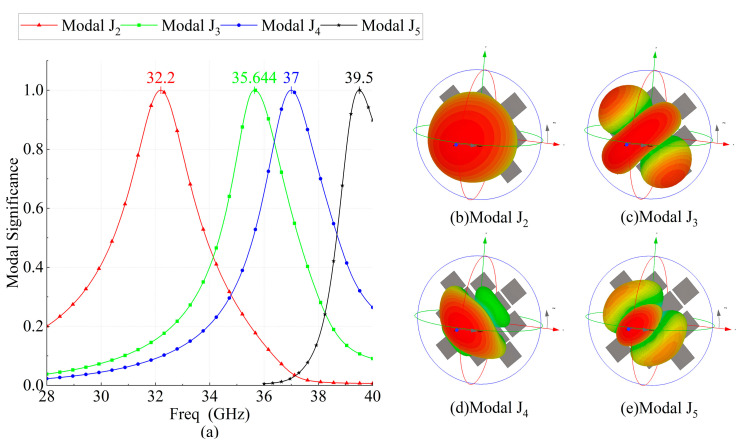
(**a**) Modal significances of the metasurface without slots; modal radiation pattern of the metasurface without slots for (**b**) Modal $J_{2}$; (**c**) Modal $J_{3}$; (**d**) Modal $J_{4}$; and (**e**) Modal $J_{5}$.

**Figure 7 micromachines-14-01403-f007:**
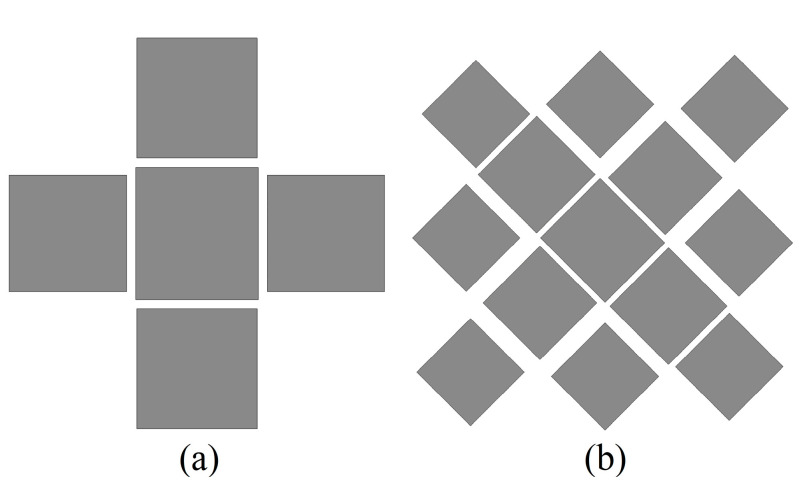
Configuration of the reference antenna for (**a**) antenna 1 and (**b**) antenna 2.

**Figure 8 micromachines-14-01403-f008:**
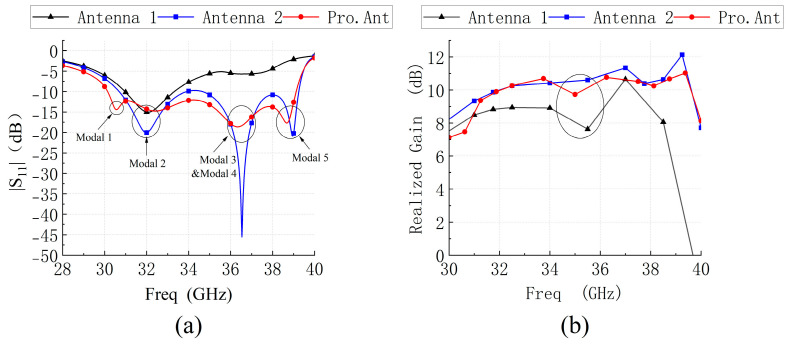
(**a**) Simulated $\left|S_{11}\right|$ and (**b**) broadside realized gains of reference antenna 1, antenna 2, and proposed antenna.

**Figure 9 micromachines-14-01403-f009:**
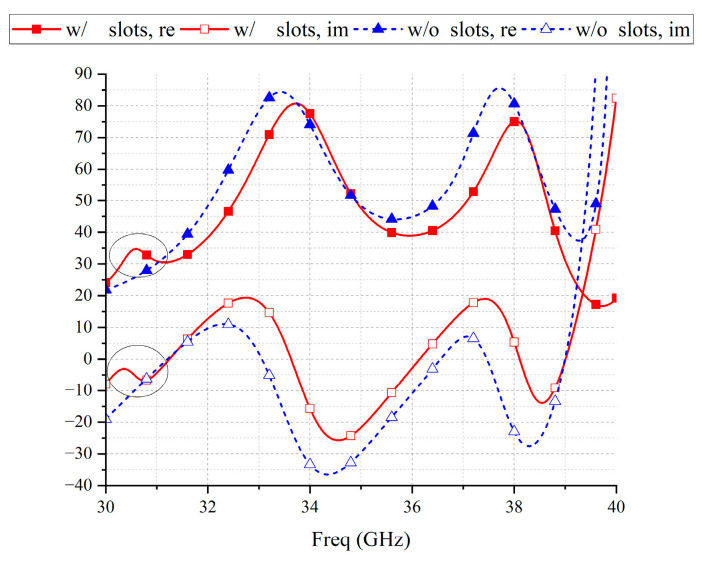
The influences of slots on impedance.

**Figure 10 micromachines-14-01403-f010:**
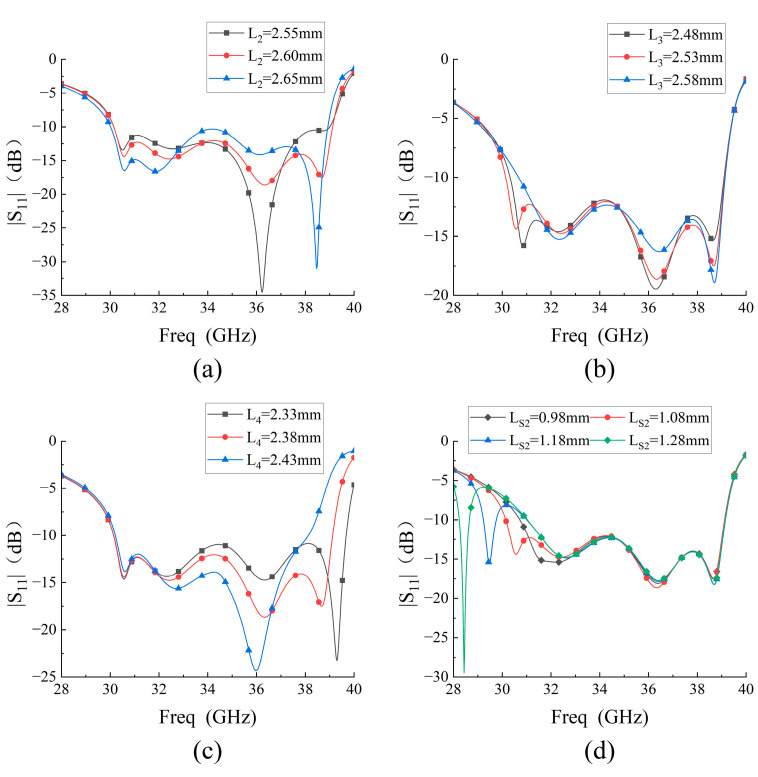
Influences of (**a**) $L_{2},\;\left(b\right)\;L_{3},\;\left(c\right)\;L_{4}$, and (**d**)
$L_{s2}$ on $\left|S_{11}\right|$.

**Figure 11 micromachines-14-01403-f011:**
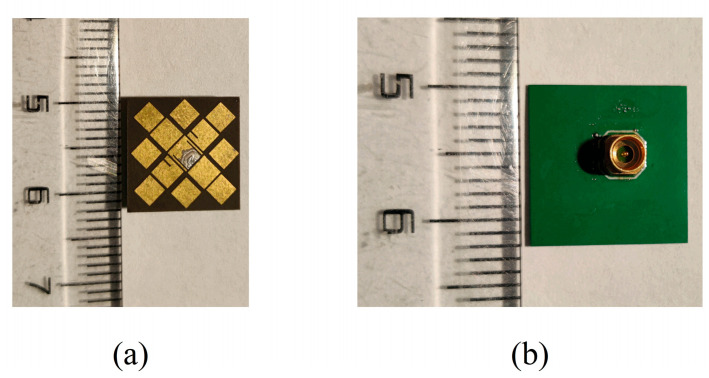
Prototype of the proposed metasurface-based antenna’s (**a**) top view and (**b**) bottom view.

**Figure 12 micromachines-14-01403-f012:**
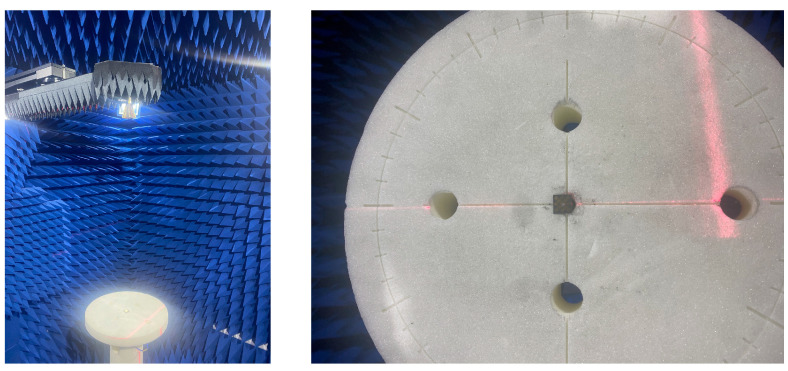
Far-field measurement environment.

**Figure 13 micromachines-14-01403-f013:**
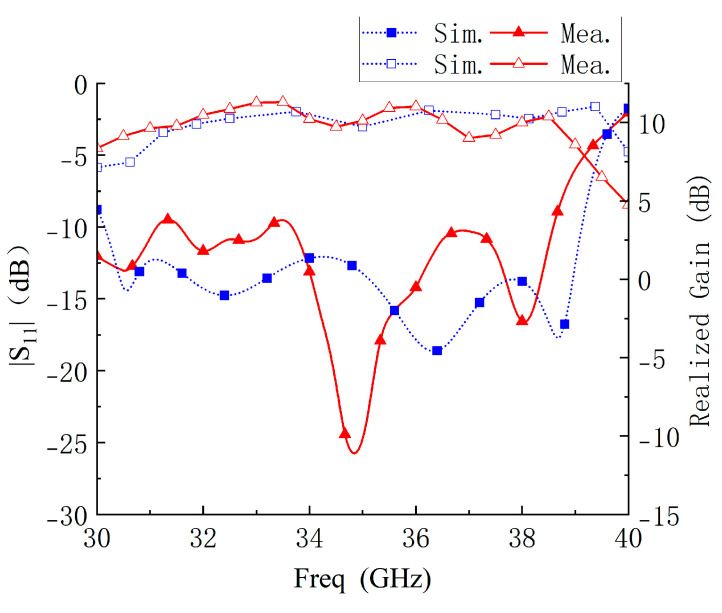
Simulated and measured $\left|S_{11}\right|$ and broadside gain.

**Figure 14 micromachines-14-01403-f014:**
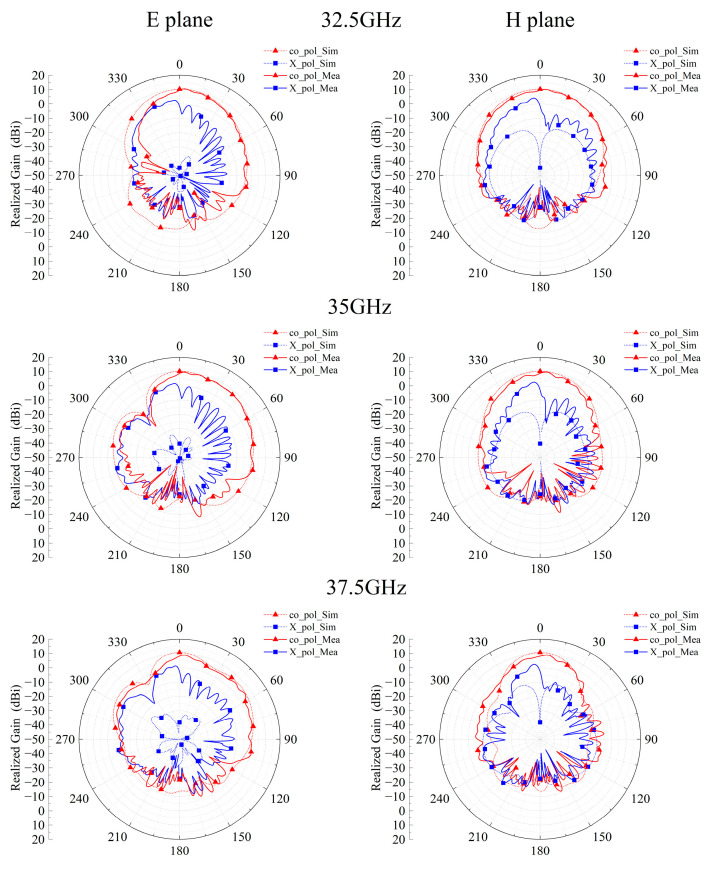
Simulated and measured radiation patterns at 32.5, 35, and 37.5 GHz.

**Figure 15 micromachines-14-01403-f015:**
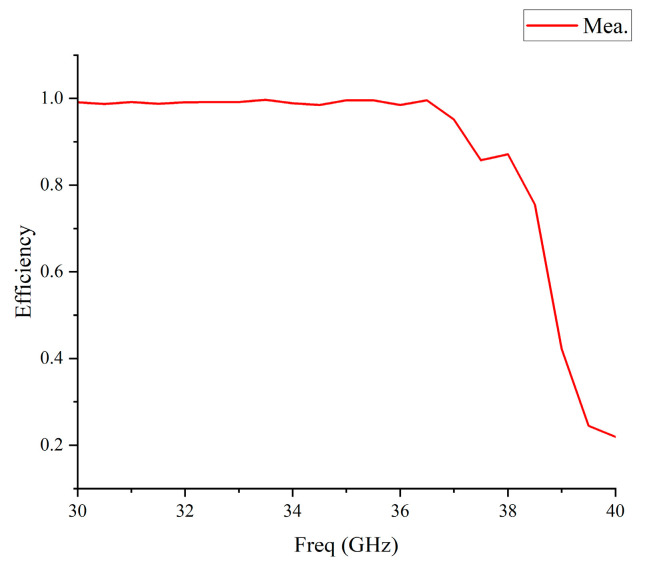
Measured radiation efficiency of the proposed antenna.

**Table 1 micromachines-14-01403-t001:** Summary of antenna geometry (unit: mm).

Par.	Val.	Par.	Val.	Par.	Val.
$W_{1}$	2.65	$S_{13}$	0.2	$L_{s1}$	2.3
$L_{1}$	2.94	$S_{14}$	0.45	$L_{s2}$	1.08
$L_{2}$	2.6	$S_{24}$	0.2	$W_{s1}$	0.15
$L_{3}$	2.53	$S_{34}$	0.3	$W_{s2}$	0.2
$L_{4}$	2.41	$D_{s1}$	2	$L_{f}$	1.2
$S_{12}$	0.2	$D_{s2}$	0.5		

## Data Availability

Data sharing not applicable.

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
