# Peer review of "Low-Profile Millimeter-Wave Metasurface-Based Antenna with Enhanced Bandwidth"

_micromachines, 2023, doi:10.3390/mi14071403_

Round 1

Reviewer 1 Report

1.      Write references in the [] brackets.

2.      The abstract and conclusion need to technically strengthen. Please rewrite it.

3.      In Table 1, are all dimensions in mm?? It is not mentioned anywhere.

4.      As per the title it is presented as bandwidth enhancement while in the paper the major discussion is only about the modal analysis. Please justify the title.

5.      Illustrate what are the manuscript’s strengths and weaknesses.

6.      What is the mathematical significance of this proposed method for enhancement?

7.      The radiation patterns have more harmonics presented. Please justify.

8.      The introduction part needs a more critical literature review. Please add a few more details.

9.      Images are not clear.

10.  References are very few. Not sufficient for justifying the work.

11.  Overall paper needs a full grammar check by an English expert.

-

Reviewer 2 Report

In this paper, a low-profile broadband metasurface based antenna with five resonance modes is proposed. Although it seems technically good, the organization and presentation of the paper is problematic and unsatisfactory. The paper needs to be handled again. The following points must be taken care of.

·         Key words should be in alphabetical order.

·         In the abstract section, the authors used a phrase such as "Using a coaxial probe to feed, and good results are obtained". The "good result" in this expression is a relatively subjective description and should be supported by comparable measurable results such as efficiency, gain, and fractional bandwidth. Also, the abstract should explain what the problem is and how it is solved here. The abstract should be well reviewed in the light of my above-mentioned assessment.

·         Subject presentation in all sections should be improved.

·         There has been a very wide range of publications about millimeter antennas recently. The novelty and originality of the study should be addressed with the introduction, which includes a comprehensive literature and related studies.

o   Sun, Q., Ban, Y. L., Li, X. F., Hu, J., & Nie, Z. (2023). A Passive Metasurface for Gain Enhancement of Wide Angle Millimeter-Wave Multibeam Array Antenna. IEEE Antennas and Wireless Propagation Letters.

o   Cao, T. N., Nguyen, M. T., Phan, H. L., Nguyen, D. D., Vu, D. L., Nguyen, T. Q. H., & Kim, J. M. (2023). Millimeter-Wave Broadband MIMO Antenna Using Metasurfaces for 5G Cellular Networks. International Journal of RF and Microwave Computer-Aided Engineering2023.

o   Das, P., & Singh, A. K. (2023). Gain Enhancement of Millimeter Wave Antenna by Ultra-thin Radial Phase Gradient Metasurface for 5G Applications. IETE Journal of Research, 1-9.

o   Saleh, C. M., Almajali, E., Jarndal, A., Yousaf, J., Alja’Afreh, S. S., & Amaya, R. E. (2023). Wideband 5G antenna gain enhancement using a compact single-layer millimeter wave metamaterial lens. IEEE Access11, 14928-14942.

o   Etc….

·         At the end of the introduction section, there should be a paragraph about the novelty of the proposed antenna design methodology (Ex: In this paper or study ….).

·         At the end of the introduction section, there should be a paragraph about the novelty of the proposed and performed work (Ex: In this paper or study ….).

·         Abstract and Conclusion Sections should be improved. The results obtained for the method used, should be included in the Abstract and Conclusion sections.

·         Page one, lines 29, 35, 36, 37 and 38, there is a problem with the reference notation. It should be shown with square brackets.

·         Patch antenna groups C1, C2, C3 and C4 identified by the authors should also be shown in Figure 1.

·         The design stages of the final metasurface antenna design process should be introduced with a new figure to be added to the paper.

·         The critical design parameters in antenna geometry in the design process and the effect of these parameters on antenna performance (such as gain, and BW) should be presented with a new "parametric study" section to be added.

·         What is the shortest spacing between two adjacent lines? What antenna fabrication technique was used to fabricate the prototype? How small a spacing between adjacent copper lines can be etched using this method?  There is no information about antenna fabrication.

·         What tool/ method (wet etching, or milling machine or other?) was followed for fabrication?

·         A picture of the antenna measurement setup should be included, and information should be given about the measurement setup (cables, VNA, anechoic chamber etc. ).

Minor editing of English language required

Reviewer 3 Report

This paper deals with a very interesting topic and achieves very promising results. However, I have some comments regarding the way some concepts are presented or introduced. In my opinion the term metasurface antenna is not quite rigorous, and the term metasurface-based antenna should be used, although it is less compact. In the design there is not as such one or several metasurfaces, because the degree of periodicity is negligible, not to say that, strictly speaking, no periodicity is kept, since resonant patches of different sizes are combined and the gap is not preserved in a regular way. A metasurface is by definition periodic and, moreover, with unit cells much smaller than the wavelength. This last detail in the microwave range is often not strictly adhered to, but the periodicity .... So, as strictly speaking I can understand that the authors have used metasurface theory as a basis for designing such structures operating at several adjacent frequencies, so that they have combined them with some degree of overlap between the bands, to achieve a much wider band than with a single band, and the central patch is the antenna whose operation is considerably improved by the unit cells and slots introduced, I would speak of a metasurface-based antenna. It is a nuance, but I think it is more rigorous, so as not to expand what I consider to be a rather widespread terminology error.

Another thing I like, however, is the explanation based on the characteristic modes. That really adds value to this contribution. Moreover, emphasis could be placed on the difference between this kind of design and the mere placement of parasitic resonant patches around the central patch. Because such a small number of unit cells are used at each frequency, some readers may wonder what the difference is between what is presented and placing metallo-dielectric resonators in the form of a square patch at different frequencies......

I am missing works on using coplanar metasurfaces to improve microstrip patch antennas, or to reduce the RCS, which has been presented by previous authors, or using characteristic modes, as for example (but not only):

R1) doi:10.1002/mop.25974 First contribution in which a patch antenna is surrounded by a metasurface in the same layer aiming to enlarge the bandwidth: https://onlinelibrary.wiley.com/doi/abs/10.1002/mop.25974

R2) doi:10.2528/PIER11040103  First contribution in combining metasurfaces with overlapped frequency bands to further reduce RCS (demonstrating that is not necessary a 180 phase different to achive so): https://www.jpier.org/PIER/pier.php?paper=11040103

R3) doi: 10.1155/2012/843754 Contribution comparing patch antenna surrounded by EBG unit-cells in the same layer, and over AMC, aiming at improving the bandwidth. Results of the radiation efficiency included: https://www.hindawi.com/journals/ijap/2012/843754/

It is not mandatory to cite them, but I think the authors should valorate if such contributions and others should be cited because the idea of surrounding a microstrip patch antenna to enhance bandwidth and the idea of combining metasurfaces with overlapped frequency bands comes from them (and not from the more recent references, which of course should be also referenced).

It is not clear for me how the CST simulation has been performed when an infinite ground plane is mentioned. Is the 13 patches structure simulated using periodic boundary conditions using an incident plane-wave, or how is the simulation set-up configured? Please clarify this point because it is important from a designer point of view.

The radiation efficiency results should be included since it is of major importance. The gain is of course important, but if the bandwidth and or the gain are improved at the cost of degrading the radiation efficiency, a reader should be aware of it. I know many authors do not provide this value, but it is mandatory to really evaluate the degree of improvement provided with this and other methods.

For example, recently the authors in R4) doi: 10.23919/EuCAP53622.2022.9769433 https://ieeexplore.ieee.org/stamp/stamp.jsp?tp=&arnumber=9769433, present a millimetre wave antenna combined with a HIS metasurface in the same layer, that provides 7.5% of bandwidth (more modest that the one provided in this contribution), but also improving Gain,directivity, front-to-back ration and keeping more than 98% radiation efficiency.

The crosspolar level rises in measurement. This is not a problem because very probably it can be explained in terms of the cable, the connector, misalignments of antennas and other issues related to the measurement set-up that are not considered in simulation. Please properly describe and show the measurement set-up and methodology to explain this point.

Finally, a minor issue related to references 1, 9-14 which are not indicated in brackets in the text.

This reviewer encourages the authors to address these comments for the paper publication since it is a valuable contribution which deserves it.

Round 2

Reviewer 2 Report

The authors have done a pretty good job when I reviewed the previous and final versions. Thank you to them.

I consider that the revised version of the manuscript is in good for publication in Micromachines journal standards.

Author Response

Thank you for your valuable advice.

Reviewer 3 Report

The authors have not addressed all the reviewer comments since part of the reviewer text has been removed or not included in the "answer to the reviewer comments"

From my point of view, there is lack of rigor when referencing where the main ideas on which they have been based come from. Early work using metasurfaces surrounding patch antennas to increase bandwidth, early work combining overlapping resonances on metasurfaces to increase bandwidth, work using characteristic modes.. the authors should answer why they are not providing references on the three main ideas the paper is based on, unless they have been the first ones...

Other parts of the paper have been correctly improved. Finally, in the title, it should be metasurface-based antenna (instead of metasurface antenna), for the same reason as in the manuscript text.
